# Metabolic Syndrome and Dietary Habits in Hospitalized Patients with Schizophrenia: A Cross-Sectional Study

**DOI:** 10.3390/medicina57030255

**Published:** 2021-03-10

**Authors:** Tamara Sorić, Mladen Mavar, Ivana Rumbak

**Affiliations:** 1Psychiatric Hospital Ugljan, Otočkih Dragovoljaca 42, 23275 Ugljan, Croatia; ravnatelj@pbu.hr; 2Faculty of Food Technology and Biotechnology, University of Zagreb, Pierottijeva 6, 10000 Zagreb, Croatia; ivana.rumbak@pbf.hr

**Keywords:** schizophrenia, metabolic syndrome, dietary habits, inpatients

## Abstract

*Background and Objectives*: The true prevalence of metabolic syndrome (MetS) and the reason for it being higher in patients with schizophrenia when compared to general population have not yet been fully determined. Although being considered as one of the major causes, currently there are limited findings regarding differences in dietary patterns of schizophrenic patients with and without MetS. The present study aimed to determine the prevalence of MetS among hospitalized patients with schizophrenia, to investigate the differences in socio-demographic, clinical, and lifestyle characteristics between participants with and without MetS, with the special emphasis being put on their dietary habits, and to ascertain the correlation between dietary habits and MetS components. *Materials and Methods*: A cross-sectional study included 259 hospitalized patients with schizophrenia aged ≥ 18 years. All participants underwent basic anthropometric measurements, blood sampling and blood pressure assessment, and provided relevant socio-demographic and lifestyle information. The presence of MetS was determined according to the Joint Interim Statement definition and dietary habits were assessed using a non-quantitative food frequency questionnaire. *Results*: The overall prevalence of MetS was 47.9%. No socio-demographic or lifestyle differences were found between participants with and without MetS. A large number of participants (42.9%) reported consuming carbonated soft drinks on a daily basis. Daily frequency of fruit (11.6%) and vegetables intake (29.3%) was far below recommended. Dietary habits of participants with and without MetS did not significantly differ, while consumption frequencies of some of the studied food and beverage items and groups significantly correlated with certain MetS components (such as statistically significant positive correlation between cured meat products consumption frequency and waist circumference, as well as between red meat consumption frequency and systolic blood pressure). *Conclusions*: The concept of the present study did not allow us to distinguish to what extent the participants’ dietary habits were influenced by independent procurement of food products, nor has it allowed us to quantify the portion sizes of consumed food and beverage items and groups. Nevertheless, the findings indicate the need for early identification of individuals with high MetS risk and for the incorporation of nutritional support programs into hospital treatment of patients with schizophrenia.

## 1. Introduction

In the last few decades, metabolic syndrome (MetS) has reached epidemic proportions in both the developed and developing countries worldwide [1]. As a set of metabolic disorders, MetS induces higher susceptibility to type 2 diabetes mellitus and cardiovascular diseases [2].

Regarding MetS prevalence, it varies depending on numerous factors including ethnicity, age, gender, and the set of diagnostic criteria used for its determination [3]. Due to the fact that several different organizations have developed their own definitions of MetS, the true prevalence on a global scale remains undetermined [4]. According to the International Diabetes Federation (IDF), MetS is present in approximately one-fourth to one-fifth of the total world’s adult population [5]. Nevertheless, as reported by Papanastasiou [6], the risk of developing MetS among patients with schizophrenia is nearly doubled when compared to the general population. Mitchell et al. [7] conducted a meta-analysis of 77 publications and reported the overall MetS prevalence of 32.5% among adults with schizophrenia and related disorders. According to the obtained results, there were just small differences in the prevalence rates depending on the definition used and whether the participants were hospitalized or not [7].

Furthermore, it is already well established that people with serious mental illnesses, including schizophrenia, are more prone to an unhealthy lifestyle when compared to the general population [8]. This fact is supported by the results of numerous previously conducted studies confirming that this specific population group has poorer dietary habits [9,10], engages in less physical activity [11], smokes more cigarettes [12], and consumes larger amounts of alcohol [13], when compared to people without mental disorders. Although the true cause for higher prevalence of MetS in individuals with schizophrenia has not yet been fully determined [14], the aforementioned factors may contribute to its development [15,16,17,18,19].

When speaking more specifically about dietary habits, the vast majority of the previously published studies compared the dietary patterns of people with schizophrenia with those of general population. According to Simonelli-Muñoz et al. [20], patients with schizophrenia have unfavorable dietary habits. Namely, 40.8% of the participants reported not eating fruit on a daily basis and 63.1% never ate fish. High incidence of unhealthy dietary habits has also been reported in the study of Heald et al. [21]. In their study, only 13.5% of outpatients with schizophrenia reported consuming recommended number of daily servings of fruit and vegetables, while 37.5% of the participants reported never eating fruit [21]. Similarly, as reported by Parletta et al. [10], people with serious mental illness in the Western countries tend to consume significantly lower amounts of high-quality foods, while unhealthy foods are consumed in higher amounts. Despite the aforementioned studies, and to the best of our knowledge, only two previous studies compared dietary habits of patients with schizophrenia with and without MetS, and found no significant differences [22,23], except for some of the applied culinary methods [23].

Therefore, the present study aimed to determine the prevalence of MetS in hospitalized patients with schizophrenia, to investigate the differences in socio-demographic, clinical, and lifestyle characteristics between participants with and without MetS, with the special emphasis being put on their dietary habits, and to ascertain the correlation between the consumption frequencies of the studied food and beverage items and groups and MetS components. We hypothesized that MetS would be established in approximately 35% of the study participants and that those participants with the diagnosis of MetS would have poorer dietary habits when compared to the ones without MetS. We have also hypothesized that significant correlations between consumption frequencies of the studied food and beverage items and groups and MetS components would be determined.

## 2. Materials and Methods

This was a cross-sectional study carried out in 2017 in Psychiatric Hospital Ugljan, a tertiary care hospital in Croatia. The present study was conducted as part of a larger study which primary aim was to investigate the impact of the Dietary Approaches to Stop Hypertension diet on MetS in this specific population group [24].

Participants’ eligibility criteria and some parts of the study protocol and methodology were published previously in the articles of the same research group that reported on the association between cereal products consumption frequency and anthropometric and biochemical parameters in hospitalized patients with schizophrenia [25], and on the main results of the randomized controlled trial [24]. For the elements already described, only a short summary will be provided in the sections below.

### 2.1. Study Participants

Patients of both genders, aged 18 years and older, that were hospitalized in the Psychiatric Hospital Ugljan at the time being and had the diagnosis of schizophrenia according to the 10th revision of the World Health Organization’s International Statistical Classification of Diseases and Related Health Problems (ICD-10), were approached by the researcher (nutritionist) to further verify their eligibility for study participation. Exclusion criteria were as follows: (a) refusal to provide all the relevant information; (b) refusal to provide written informed consent for study participation either from the participant personally or respective legal guardian; (c) cognitive or physical deteriorations that could disable collection of all the relevant information and participant’s ability to fully participate in the study. The recruitment lasted until a sufficient number of participants in both groups (with and without MetS) was gathered.

### 2.2. Ethics

All the participants engaged in the study on a voluntary basis and had to provide written informed consent after all the aims and study procedures had been explained in detail. For those participants who were deprived of legal capacity, written informed consents were provided from both the participants and their legal guardians. Seeking consents from legal guardians has been demanded by the ethics committee as it is a legal requirement in Croatia. Those participants who have accepted to take part in the present study, but their legal guardians refused to sign the informed consent, and vice versa, were not included. The study was approved by the Ethics Committee of the Psychiatric Hospital Ugljan (protocol code: 01-552/01-16; date of approval: 21 November 2016) and the Central Ethics Committee of the Medical School, University of Zagreb (protocol code: 380-59-10106-17-100/56; date of approval: 23 February 2017). The study was conducted in accordance with the Declaration of Helsinki.

### 2.3. Measurements and Assessments

Individual face-to-face interviews were undertaken by the researcher (nutritionist) to gather all the relevant data from the study participants, including socio-demographic details (age, educational level, working status, marital status, residential area) and lifestyle information (smoking status, alcohol consumption, physical activity level, sleeping pattern). Participants’ psychiatric history details were provided by the medical personnel of the departments where participants were treated at the time of their participation in the present study.

Anthropometric measurements were performed by the researcher (nutritionist) in accordance with the standard instructions [26]. Body height (BH) and body weight (BW) were measured barefoot and in light clothing. Afterwards, body mass index (BMI) was calculated as the ratio of the participant’s BW in kilograms and BH in meters squared. The measurements of the waist circumference (WC) and hip circumference (HC) were performed while participants were standing in the relaxed position and with arms at the side. The waist-to-hip ratio (WHR) was calculated from the obtained values according to the following formula: WC (cm)/HC (cm). BW was measured to the nearest 0.1 kg, while BH, WC, and HC were recorded to the nearest 0.1 cm. Percentage of body fat (% BF) was determined by bioelectric impedance analysis. Values of BW and % BF were used for the calculation of each participant’s BF mass in kilograms.

Participants were in a sitting position for at least five minutes prior to blood pressure (BP) assessment. In total, two trained nurses performed two BP measurements for each participant (the same nurse performed both measurements on an individual participant) with a five-minute interval between measurements as stated by Frese et al. [27]. For research purposes, each participant’s BP mean value calculated from the two measurements was used.

Furthermore, blood serum concentrations of total cholesterol (TC), high-density lipoprotein cholesterol (HDL-C), low-density lipoprotein cholesterol (LDL-C), triglycerides (TG), and glucose (GLC) were determined by the trained laboratory personnel. Concentrations of TC, HDL-C, TG, and GLC were determined enzymatically, while LDL-C concentration was calculated from the obtained concentrations of TC, TG, and HDL-C following the Fiedewald’s formula [28]. Blood samples from all the participants were collected after a 12-h overnight fast.

After all the necessary data have been collected, participants were screened to verify if they meet the criteria for MetS according to the Joint Interim Statement (JIS) definition. According to the definition, participants were considered to have MetS if they fulfilled at least three of a total of five criteria listed in Table 1.

Participants’ dietary habits were assessed in a form of face-to-face interviews using a slightly modified part of the nutrition section of Dlugosch and Krieger’s General Health Behavior Questionnaire [30]. According to Jakabek et al. [31], this questionnaire is considered as an appropriate tool to evaluate nutrition behavior in people with serious mental illness, including people with the diagnosis of schizophrenia. The translation and validation process of the questionnaire was described in our previous article [25]. A modified part of the nutrition section of the aforementioned questionnaire represents a non-quantitative food frequency questionnaire consisting of 32 items (25 food items and groups and 7 beverage items and groups). The consumption frequency of each food and beverage item and group was estimated based on the choice of one response on a four-point Likert scale. The four consumption frequencies available for food items and groups were: (1) every day; (2) few times a week; (3) rarely; and (4) never. For beverage items and groups participants were able to choose between the following responses: (1) several times a day; (2) once a day; (3) rarely; and (4) never. The reference period for dietary habits assessment was participant’s last three months.

### 2.4. Statistical Analysis

Statistical software used for the statistical analysis in the present study was Statistica 6.1 (StatSoft Inc., Tulsa, OK, USA).

The needed sample size was calculated before the study start using module Power Analysis—Sample Size Calculation of the aforementioned statistical software [32]. The sample size calculation was based on the two main outcomes of the present study. According to the power analysis, a sample size of 83 participants was needed to achieve 95% confidence level with the level for statistical significance of 0.05, a power goal of 0.90, and under the null hypothesis that the established MetS prevalence among hospitalized patients with schizophrenia will be higher than the estimated prevalence in the general population (35 and 20%, respectively). Additionally, a sample size of 246 participants (*n* ≥ 123 with MetS and *n* ≥ 123 without MetS) was needed to achieve 95% confidence level with the level for statistical significance of 0.05, average standard deviation (SD) of the answers to questions in the non-quantitative food frequency questionnaire σ = 1.00, standardized effect −0.415, a power goal of 0.90, and under the null hypothesis of MetS presence being associated with poorer dietary habits.

Normality of the distribution was tested by Kolmogorov–Smirnov test and Shapiro–Wilk W-test. Socio-demographic, clinical, and lifestyle characteristics of the study participants were presented as means and SDs for numerical data, and as absolute frequencies and percentages for categorical data. Anthropometric characteristics, BP, and biochemical parameters were presented as means and SDs, together with 95% confidence intervals. Consumption frequencies of all the studied food and beverage items and groups were presented as medians and interquartile ranges, as well as absolute frequencies and percentages. For the evaluation of differences in socio-demographic, clinical, and lifestyle characteristics between the participants with and without MetS, the independent samples t-test and Mann–Whitney U test were used for numerical data, and the Pearson Chi-square test and Mann–Whitney U test for categorical data. The values of anthropometric and biochemical parameters and BP were compared between the groups using the independent samples t-test. The Mann–Whitney U test was used to compare consumption frequencies of all the studied food and beverage items and groups between the participants with and without MetS. The associations between the consumption frequencies of all studied food and beverage items and groups and MetS components were assessed using Spearman’s rank correlation coefficients. All *p*-values below 0.05 were observed as statistically significant.

## 3. Results

In total, 259 participants completed the study. The unequal gender distribution of the study participants was a direct result of a standard proportion of male and female patients hospitalized in Psychiatric Hospital Ugljan (approximately 75% male patients and 25% female patients).

A total of 124 participants (47.9%) met at least three of the overall five criteria and were therefore diagnosed with MetS. Among the participants with MetS, 58 (46.8%) had three, 52 (41.9%) had four, and 14 (11.3%) had five positive components for the determination of MetS. Only 21 out of 259 participants (8.1%) had no positive components for MetS. The number and proportion of participants with and without MetS meeting the individual criteria of a JIS definition is presented in Table 2.

Table 3 summarizes basic characteristics of the study participants divided into two groups, depending on the presence of MetS. The majority of the participants had a high school education (60.6%), were retired (57.1%), and were single (91.1%). As for the lifestyle characteristics, the vast majority of the participants were physically inactive (66.4%) and smokers (71.8%). There were no statistically significant differences between participants with and without MetS in relation to any of the studied socio-demographic, clinical, and lifestyle characteristics listed in Table 3.

Relevant anthropometric characteristics, BP measures, and biochemical parameters of the participants with and without MetS are presented in Appendix A. The values of all studied anthropometric parameters, except for BH (*p* = 0.395) were significantly higher in participants with MetS, when compared to those without MetS (all *p* < 0.001). Participants with MetS also had significantly higher systolic blood pressure (SBP) and diastolic blood pressure (DBP) (both *p* < 0.001), together with significantly higher values of TC (*p* = 0.009), LDL-C (*p* = 0.039), TG (*p* < 0.001), and GLC (*p* < 0.001), and significantly lower values of HDL-C (*p* < 0.001), when compared to the participants without MetS (Appendix A).

In terms of dietary habits, for the study participants the daily frequency of fruit (11.6%) and vegetables intake (29.3%) was far below recommended. Cured meat products were consumed few times a week by 149 participants (57.5%). A vast majority of the participants reported never consuming nuts (76.1%). Regarding the consumption frequency of beverage items and groups, a large number of participants (42.9%) reported consuming carbonated soft drinks on a daily basis. Appendix A show the frequencies of answers to individual questions that formed part of a non-quantitative food frequency questionnaire for both the participants with and without MetS. No statistically significant differences in dietary habits were found between participants with and without MetS (Table 4).

Lastly, we have assessed the relationship between consumption frequencies of all studied food and beverage items and groups and those parameters that form a fundamental part of the MetS concept (Table 5). The consumption frequency of cured meat products and fruit juice and lemonade positively correlated with WC (*p* = 0.031 and *p* = 0.001, respectively). Similarly, the frequency of red meat intake positively correlated with SBP and DBP (*p* = 0.040 and *p* = 0.041, respectively), while poultry and carbonated mineral water consumption frequencies positively correlated only with SBP (*p* = 0.033 and *p* = 0.003, respectively). As for the coffee with caffeine and black tea, a statistically significant positive correlation of their intake frequency and DBP and GLC (*p* = 0.037 and *p* = 0.043, respectively) was found. The concentration of TG positively correlated with the intake frequency of bread and bagels made of rye or whole-wheat flour (*p* = 0.018) and butter and margarine (*p* = 0.029), and negatively with the intake frequency of eggs (*p* = 0.044). The consumption frequencies of breakfast cereals and muesli (*p* = 0.036), cakes (*p* = 0.039), cured meat products (*p* = 0.009), and potato (*p* = 0.049) negatively correlated with HDL-C concentration. A statistically significant negative correlation was observed between the intake frequency of jam and honey and GLC level (*p* = 0.021), as well as WC (*p* = 0.047).

## 4. Discussion

The present observational study, which had a cross-sectional design, demonstrated relatively high prevalence of MetS among hospitalized individuals with the diagnosis of schizophrenia and it identified inadequate dietary habits that did not differ significantly between the participants with and without MetS. In addition, some of the studied food and beverage items and groups significantly correlated with MetS components.

Although it was expected that MetS would be identified in a relatively large number of participants, the prevalence established in the present study was surprisingly high with almost half of the study participants meeting three or more criteria required for the determination of MetS according to the JIS definition. Therefore, the determined MetS prevalence exceeded an average prevalence rates from previously published meta-analyses conducted on patients with schizophrenia [7,33,34,35]. In few more recent studies, similarly to the aforementioned, the prevalence of MetS among inpatients with schizophrenia varied between 30.6 and 37.8% [36,37,38]. However, it is important to emphasize that the JIS definition has not been used in the above-mentioned studies, which consequently prevents a direct comparison with the results of this research [2]. The studies, conducted on different population groups, that were focused on the comparison of MetS prevalence when using different definitions reported the highest prevalence rate with JIS definition [39,40,41]. The difference between the JIS definition and the two most commonly used MetS definitions lies in the criteria for abdominal obesity, whereas all the remaining features are essentially identical. According to the IDF definition, abdominal obesity is a precondition for the diagnosis of MetS [29]. Same as in the JIS definition, in the modified National Cholesterol Education Program Adult Treatment Panel III (NCEP ATP III), abdominal obesity is one of five criteria, out of which it is necessary to have at least three for the diagnosis [29]. However, the proposed cut-off values for WC in the modified NCEP ATP III definition are higher in both men and women (≥102 cm and ≥88 cm, respectively) when compared to the IDF and JIS definitions (for Europids: ≥94 cm and ≥80 cm, respectively) [29]. When observing the aforementioned, it can be concluded that, on the basis of the abdominal obesity criteria, people are more likely to have MetS with JIS definition than if the IDF or modified NCEP ATP III definitions are applied. Therefore, even though the study concept did not allow us to conclude on the exact reasons for high prevalence of MetS among hospitalized patients with schizophrenia in the present study, the use of JIS definition could be the reason for a higher prevalence rate than hypothesized. Since MetS is being tightly associated with the increased risk of developing cardiovascular diseases and type 2 diabetes mellitus [2], the importance of this result lies in the fact that it represents a strong evidence of MetS problem in this specific population group.

When looking at the individual components of MetS, the results of the present study confirm that abdominal obesity is the most prevalent parameter in the diagnosis of MetS [42].

Numerous previously conducted studies focused on socio-demographic, clinical, and lifestyle predictors of MetS in patients with the diagnosis of schizophrenia, however until today none of the studied parameters has arisen as a permanent predictor of MetS in this specific population group and consequently there is still no indisputable evidence regarding the connection between these variables and MetS [43]. The results of the study at hand support the aforementioned fact, as we have found no significant differences between participants with and without MetS, considering all the studied socio-demographic, clinical, and lifestyle characteristics (Table 3). For the interpretation of the results it is important to emphasize that in the present study we have observed the self-reported physical activity level, but did not account for its type and duration. As both the type and duration of the performed activity are known to be closely related to MetS [44,45], the lack of observance thereof could possibly explain the absence of the association between physical activity and MetS in the present study. Furthermore, the use of antipsychotics, especially atypical antipsychotics, significantly increases the risk of MetS [46]. The present study took into account the type and number of different antipsychotics used, but did not observe the antipsychotic doses. As there was no statistically significant difference between the participants with and without MetS regarding the type and number of antipsychotics used, the relationship between MetS and antipsychotics could potentially be dose-dependent. However, the present study did not allow us to conclude on a relationship between MetS and applied antipsychotic doses. The results of the study at hand also indicate that it is probably not only for the socio-demographic, clinical, and lifestyle contributors towards MetS, but that the high rates of MetS in people with schizophrenia might likely be influenced by various genetic factors and biological markers [43,47].

Despite the average age of both the participants with and without MetS (52.5 and 52.3 years, respectively) more than half of the study participants were retired. The obtained result is not surprising when being observed in the context of mental illness, as schizophrenia is the predominant reason for early retirement due to illness severity [48].

A slightly modified part of the nutrition section of Dlugosch and Krieger’s General Health Behavior Questionnaire did not allow us to estimate the serving sizes, in addition to the consumption frequencies of respective food and beverage items and groups. Notwithstanding the above, it can be concluded that certain dietary habits of hospitalized patients with schizophrenia are inadequate when the obtained results are compared to the Dietary Guidelines for Adults in Croatia [49]. The established dietary habits were mainly characterized by a high consumption frequency of cured meat products and carbonated soft drinks. The majority of the study participants reported insufficient consumption frequency of fruits, vegetables, nuts, and fish. Considering cereal products consumption, the results of our recently published study showed that 80.3% of hospitalized patients with schizophrenia consumed wheat or mixed-wheat bread and bagels on a daily basis, while only 2.3% of the studied population group consumed rye or whole-wheat bread and bagels equally frequent [25]. Therefore, our results are fully consistent with those of the previously conducted studies that have shown poor dietary habits among patients with schizophrenia [10,20,21]. The vast majority of the available studies compared dietary habits of patients with schizophrenia to the general population; while only two, according to our knowledge, examined the differences in their dietary habits depending on MetS presence [22,23]. In the mentioned studies, and similar to our findings, dietary habits of those participants with and without MetS were very similar, even though it was assumed that they would significantly differ. Therefore, despite the fact that inadequate eating patterns represent one of the key risk factors for the development of MetS [16], our second hypothesis that hospitalized schizophrenic patients with MetS would have poorer dietary habits when compared to the individuals without MetS was refuted.

Finally, in the present study certain food and beverage items and groups significantly correlated with MetS components. Beside some predictable correlations, certain established associations, including a significant positive association between the consumption frequency of bread and bagels made from rye or whole-wheat flour and TG, were unexpected. Similarly to the results of the present study, Kalinowska et al. [50] have also reported on numerous significant correlations between dietary habits and biochemical parameters in patients with schizophrenia. However, when interpreting the results of the study at hand, it is of an utmost importance to take into account the fact that those dietary habits were accessed using a non-quantitative food frequency questionnaire which did not estimate the portion sizes of the consumed items. Despite the aforementioned, the very fact that certain food and beverage items and groups significantly correlated with MetS components confirms the tight connection between diet and health.

Due to the fact that conducting research on people with serious mental illnesses, including schizophrenia, is extremely challenging, and with the aim to minimize the potential recall bias, participants’ dietary habits were assessed in a form of a face-to-face interviews. With the aim to maximally reduce research participation burden, a non-quantitative food frequency questionnaire was used as a tool for dietary habits assessment.

The results of the present study should be considered in the context of several limitations. Even though the present study was conducted in an inpatient setting, meaning that it included solely hospitalized patients with schizophrenia, it is noteworthy that the established dietary habits are not exclusively a reflection of the diet provided by the Psychiatric Hospital Ugljan, but are greatly influenced by participants’ personal choices. As pointed out in our previous study [24], during hospitalization it is not possible to restrain the independent procurement of food products, which ultimately affects the overall dietary pattern of the participants. It is important to highlight that the factors, such as the deprivation of legal capacity and the type of ward in which participants were located during their hospital stay, did not impact the participants’ ability to individually purchase and consume food products, in addition to the meals provided by the Psychiatric Hospital Ugljan. Unfortunately, the concept of the present study and the methods used did not allow us to distinguish to what extent the participants’ dietary habits were influenced by self-procurement of food products. Additionally, the food frequency questionnaire used for the assessment of dietary habits did not quantify the portion sizes. The aforementioned could be seen as a limitation of the present study since it could lead to the imprecise classification of a diet and, consequently, have an impact on the association between the identified food and beverage items and groups consumption frequencies and MetS components. Another limitation of the study is not collecting data on participants’ original socio-economic background and not evaluating its potential impact and correlation to MetS and participants’ dietary habits.

The present study adds to a body of evidence that MetS is highly prevalent in hospitalized patients with schizophrenia, as well as that inadequate dietary habits are common in this specific population group. The established associations of certain food and beverage items and groups consumption frequencies and MetS components indicate a need for the incorporation of nutritional support programs into a psychiatric health care system which could result in the improvements of patients’ dietary habits and the overall quality of life. Further large prospective cohort studies and randomized controlled trials focused on nutritional interventions and diet modifications are needed to elucidate the relationship between schizophrenia, dietary habits, and MetS. Future studies should also consider the use of a quantitative food frequency questionnaire and 24-h dietary recalls for the assessment of dietary intake to collect as detailed data as possible. The application of these dietary assessment methods would also allow us to determine the degree of deviation of participants’ dietary habits in relation to the meals provided by the hospital.

## 5. Conclusions

The present study demonstrated high prevalence of MetS among hospitalized patients with schizophrenia and identified some inadequate dietary habits that did not significantly differ between hospitalized schizophrenic patients with and without MetS. Established associations between certain food and beverage items and groups consumption frequencies and MetS components in hospitalized patients with schizophrenia help understand the complex relationship between schizophrenia, dietary habits, and MetS.

## Figures and Tables

**Table 1 medicina-57-00255-t001:** Diagnostic criteria of a Joint Interim Statement definition [29].

Parameter	Cut-Off Points
WC	for Europids: ≥94 cm (men); ≥80 cm (women)
TG	≥1.7 mmol/L or pharmacological therapy for increased TG
HDL-C	< 1.0 mmol/L (men); < 1.3 mmol/L (women) or pharmacological therapy for decreased HDL-C
BP	SBP ≥130 mmHg and/or DBP ≥85 mmHg or pharmacological therapy in patients with the diagnosis of hypertension
GLC	≥5.6 mmol/L or pharmacological therapy for increased GLC

WC, waist circumference; TG, triglycerides; HDL-C, high-density lipoprotein cholesterol; BP, blood pressure; SBP, systolic blood pressure; DBP, diastolic blood pressure; GLC, glucose.

**Table 2 medicina-57-00255-t002:** The number and proportion of participants with and without metabolic syndrome meeting the individual Joint Interim Statement definition criteria.

Criteria	With MetS (*n* = 124) *n* (%)	Without MetS (*n* = 135) *n* (%)
WC	120 (96.8)	76 (56.3)
TG	78 (62.9)	14 (10.4)
HDL-C	91 (73.4)	24 (17.8)
BP	90 (72.6)	45 (33.3)
GLC	73 (58.9)	17 (12.6)

MetS, metabolic syndrome; n, number of participants; WC, waist circumference; TG, triglycerides; HDL-C, high-density lipoprotein cholesterol; BP, blood pressure; GLC, glucose.

**Table 3 medicina-57-00255-t003:** Socio-demographic, clinical, and lifestyle characteristics of the study participants.

Characteristics	With MetS (*n* = 124)	Without MetS (*n* = 135)	*p*
	*n* (%) or Mean ± SD	*n* (%) or Mean ± SD	
Sex			
male	98 (79.0)	111 (82.2)	0.515 ^c^
female	26 (21.0)	24 (17.8)	
Age (years)	52.5 ± 8.6	52.3 ± 9.7	0.836 ^d^
Educational level			
no formal education	7 (5.6)	9 (6.7)	
primary education	24 (19.4)	44 (32.6)	0.158 ^c^
secondary education	86 (69.4)	71 (52.6)	
higher educational level	7 (5.6)	11 (8.1)	
Working status			
employed	1 (0.8) ^f^	0 (0.0) ^f^	
unemployed	44 (35.5)	41 (30.4)	0.542 ^c^
retired	69 (55.6)	79 (58.5)	
receiver of social welfare	10 (8.1)	15 (11.1)	
Marital status			
single	110 (88.7)	126 (93.3)	0.212 ^c^
married or in a relationship	14 (11.3)	9 (6.7)	
Residential area			
urban	62 (50.0)	65 (48.1)	0.765 ^c^
rural	62 (50.0)	70 (51.9)	
Type of schizophrenia			
paranoid	93 (75.0)	103 (76.3)	0.829 ^c^
residual	27 (21.8)	28 (20.7)	
other	4 (3.2)	4 (3.0)	
Illness duration (years)	16.2 ± 12.2	17.2 ± 13.7	0.765 ^e^
Number of hospitalizations ^a^	7.4 ± 8.4	7.9 ± 10.1	0.497 ^e^
Duration of a current hospitalization ^a^ (years)	2.7 ± 5.5	3.1 ± 5.3	0.845 ^e^
Number of antipsychotics			
1	34 (27.4)	30 (22.2)	
2–3	84 (67.7)	94 (69.6)	0.142 ^e^
>3	6 (4.8)	11 (8.1)	
Type of antipsychotics			
typical	5 (4.0)	10 (7.4)	
atypical	65 (52.4)	60 (44.4)	0.293 ^c^
typical + atypical	54 (43.6)	65 (48.2)	
Deprivation of legal capacity			
yes	33 (26.6)	47 (34.8)	0.153 ^c^
no	91 (73.4)	88 (65.2)	
Physical activity			
inactive	86 (69.4)	86 (63.7)	
occasional activity	35 (28.2)	40 (29.6)	0.167 ^c^
frequent activity	3 (2.4)	9 (6.7)	
Smoking status			
current smokers	93 (75.0)	93 (68.9)	0.275 ^c^
former smokers and never smokers	31 (25.0)	42 (31.1)	
Number of cigarettes per day ^b^			
<10	17 (18.3)	18 (19.4)	
10–20	19 (20.4)	19 (20.4)	0.973 ^c^
>20	57 (61.3)	56 (60.2)	
Alcohol consumption			
yes	14 (11.3)	23 (17.0)	0.187 ^c^
no	110 (88.7)	112 (83.0)	
Number of sleeping hours per night			
<8	33 (26.6)	43 (31.9)	
8–10	68 (54.8)	64 (47.4)	0.481 ^c^
>10	23 (18.5)	28 (20.7)	

MetS, metabolic syndrome; n, number of participants; SD, standard deviation; ^a^ number of hospitalizations and duration of current hospitalization in Psychiatric Hospital Ugljan; ^b^ the calculation was made based on the number of participants who reported to be current smokers; ^c^ Pearson Chi-square test; ^d^ independent samples t-test; ^e^ Mann-Whitney U test; ^f^ values excluded from statistical analysis. Statistically significant: *p* < 0.05.

**Table 4 medicina-57-00255-t004:** Consumption frequency of studied food and beverage items and groups in study participants.

Parameter	With MetS (*n* = 124) Median (IQR)	Without MetS (*n* = 135) Median (IQR)	*p* ^a^
bread and bagels: wheat or mixed wheat flour	1.0 (1.0–1.0)	1.0 (1.0–1.0)	0.696
bread and bagels: rye or whole-wheat flour	4.0 (3.0–4.0)	4.0 (4.0–4.0)	0.098
burek, puff-pastry, donuts, strudels, and similar bakery products	3.0 (2.0–3.0)	3.0 (2.0–4.0)	0.534
breakfast cereals, muesli	3.0 (3.0–3.0)	3.0 (3.0–4.0)	0.551
cakes	3.0 (2.0–3.0)	3.0 (2.0–3.0)	0.975
butter, margarine	2.0 (2.0–2.0)	2.0 (2.0–3.0)	0.055
eggs	2.0 (2.0–3.0)	2.0 (2.0–3.0)	0.064
jam, honey	2.0 (2.0–3.0)	2.0 (2.0–2.0)	0.066
cured meat products (sausages, hot-dogs, salami, prosciutto, budjola)	2.0 (2.0–3.0)	2.0 (2.0–3.0)	0.308
low-fat milk and dairy products	2.0 (1.0–4.0)	2.0 (2.0–4.0)	0.106
whole milk and dairy products	2.0 (2.0–3.0)	2.0 (2.0–3.0)	0.635
semi-hard cheese (e.g., Emmental cheese)	2.0 (2.0–3.0)	2.0 (2.0–3.0)	0.848
fruits	2.0 (2.0–3.0)	2.0 (2.0–3.0)	0.681
chocolate, candies, cookies, pudding	2.0 (2.0–3.0)	3.0 (2.0–3.0)	0.181
nuts (walnuts, hazelnuts, almonds)	4.0 (3.0–4.0)	4.0 (4.0–4.0)	0.322
snacks (salted sticks, potato chips, flips)	3.0 (2.0–4.0)	3.0 (2.0–4.0)	0.422
potato	2.0 (2.0–2.0)	2.0 (2.0–2.0)	0.121
pasta	2.0 (2.0–2.0)	2.0 (2.0–2.0)	0.832
rice	2.0 (2.0–2.0)	2.0 (2.0–2.0)	0.629
vegetables	2.0 (1.0–2.0)	2.0 (1.0–2.0)	0.511
red meat (e.g., beef, pork)	2.0 (2.0–2.0)	2.0 (2.0–2.0)	0.604
poultry (chicken, turkey)	2.0 (2.0–2.0)	2.0 (2.0–2.0)	0.638
fish (including clams and mollusks)	3.0 (3.0–3.0)	3.0 (3.0–3.0)	0.126
fast food (e.g., burgers, French fries)	4.0 (3.0–4.0)	4.0 (3.0–4.0)	0.206
semi-prepared and prepared food (instant, deep frozen, canned)	2.0 (2.0–3.0)	2.0 (2.0–3.0)	0.201
coffee with caffeine, black tea	1.0 (1.0–2.0)	2.0 (1.0–3.0)	0.243
fruit juice, lemonade	3.0 (2.0–3.0)	3.0 (2.0–3.0)	0.118
herbal tea, fruit tea	2.0 (1.0–3.0)	2.0 (1.0–3.0)	0.874
hot cocoa, hot chocolate	3.0 (3.0–4.0)	3.0 (3.0–3.0)	0.207
carbonated soft drinks	3.0 (1.0–4.0)	3.0 (2.0–3.0)	0.884
light drinks (with reduced energy value)	4.0 (4.0–4.0)	4.0 (4.0–4.0)	0.251
carbonated mineral water	3.0 (3.0–4.0)	3.0 (3.0–4.0)	0.539

MetS, metabolic syndrome; n, number of participants; IQR, interquartile range; ^a^ Mann–Whitney U test. Statistically significant: *p* < 0.05. Consumption frequency data for all studied food and beverage items and groups was gathered with the use of a 1 to 4 Likert scale (Food items and groups: 1—every day, 2—few times a week, 3—rarely, 4—never; Beverage items and groups: 1—few times a day, 2—once a day, 3—rarely, 4—never).

**Table 5 medicina-57-00255-t005:** Associations between consumption frequencies of studied food and beverage items and groups and metabolic syndrome components.

Parameter	WC		SBP		DBP		TG		HDL-C		GLC	
	*r_s_*	*p*	*r_s_*	*p*	*r_s_*	*p*	*r_s_*	*p*	*r_s_*	*p*	*r_s_*	*p*
bread and bagels: wheat or mixed wheat flour	0.063	0.311	0.048	0.439	0.009	0.881	0.001	0.986	−0.012	0.852	0.057	0.358
bread and bagels: rye or whole-wheat flour	0.047	0.451	−0.007	0.907	−0.053	0.394	0.147	**0.018**	−0.040	0.521	0.023	0.708
burek, puff-pastry, donuts, strudels, and similar bakery products	0.075	0.230	0.065	0.295	0.077	0.218	0.057	0.360	−0.073	0.239	0.085	0.173
breakfast cereals, muesli	−0.003	0.961	−0.017	0.790	0.012	0.850	0.008	0.904	−0.130	**0.036**	−0.026	0.682
cakes	0.046	0.458	0.086	0.168	0.097	0.118	0.074	0.237	−0.128	**0.039**	0.031	0.616
butter, margarine	0.076	0.225	0.032	0.605	0.019	0.763	0.136	**0.029**	−0.035	0.579	0.058	0.349
eggs	−0.040	0.518	0.018	0.775	−0.037	0.548	−0.125	**0.044**	0.002	0.978	−0.068	0.277
jam, honey	−0.124	**0.047**	−0.121	0.052	−0.073	0.240	−0.018	0.774	−0.036	0.564	−0.143	**0.021**
cured meat products (sausages, hot-dogs, salami, prosciutto, budjola)	0.134	**0.031**	−0.035	0.577	−0.015	0.812	0.009	0.880	−0.161	**0.009**	0.078	0.208
low-fat milk and dairy products	0.073	0.241	0.077	0.217	0.034	0.585	0.102	0.102	0.048	0.440	0.014	0.822
whole milk and dairy products	0.073	0.240	0.035	0.573	0.068	0.273	0.035	0.580	−0.034	0.591	−0.017	0.781
semi-hard cheese (e.g., Emmental cheese)	0.103	0.097	−0.011	0.861	−0.034	0.589	0.023	0.717	−0.048	0.446	0.013	0.837
fruits	0.033	0.597	0.045	0.467	0.090	0.148	0.076	0.224	0.048	0.443	0.028	0.659
chocolate, candies, cookies, pudding	0.058	0.352	−0.081	0.195	−0.028	0.659	0.086	0.168	−0.110	0.079	0.104	0.095
nuts (walnuts, hazelnuts, almonds)	0.038	0.545	0.035	0.573	0.058	0.349	0.120	0.055	−0.054	0.386	0.024	0.700
snacks (salted sticks, potato chips, flips)	0.059	0.347	0.112	0.073	0.105	0.093	0.035	0.571	−0.085	0.172	0.004	0.952
potato	0.047	0.452	0.022	0.730	0.020	0.748	0.003	0.966	−0.123	**0.049**	0.072	0.247
pasta	0.006	0.925	−0.030	0.633	0.034	0.581	−0.027	0.667	−0.064	0.306	−0.018	0.768
rice	−0.089	0.153	−0.032	0.604	−0.035	0.579	−0.068	0.276	0.001	0.984	−0.117	0.060
vegetables	−0.004	0.948	−0.030	0.629	−0.051	0.416	−0.025	0.693	0.041	0.507	0.011	0.866
red meat (e.g., beef, pork)	0.000	0.995	0.128	**0.040**	0.127	**0.041**	0.004	0.947	0.074	0.236	−0.033	0.600
poultry (chicken, turkey)	0.056	0.369	0.132	**0.033**	0.104	0.096	0.027	0.670	−0.003	0.958	0.010	0.872
fish (including clams and mollusks)	0.059	0.341	0.044	0.477	−0.038	0.548	0.030	0.631	−0.113	0.070	0.065	0.299
fast food (e.g., burgers, French fries)	0.061	0.325	0.107	0.086	0.050	0.421	0.110	0.078	−0.086	0.169	−0.023	0.709
semi-prepared and prepared food (instant, deep frozen, canned)	0.042	0.501	0.045	0.466	0.044	0.484	0.059	0.346	−0.014	0.826	−0.098	0.115
coffee with caffeine, black tea	0.066	0.290	0.114	0.068	0.130	**0.037**	0.121	0.052	−0.054	0.386	0.126	**0.043**
fruit juice, lemonade	0.199	**0.001**	−0.011	0.861	−0.012	0.853	0.064	0.302	−0.086	0.169	0.081	0.192
herbal tea, fruit tea	−0.017	0.780	−0.042	0.504	−0.027	0.662	−0.004	0.954	−0.051	0.416	0.009	0.887
hot cocoa, hot chocolate	0.021	0.733	−0.014	0.826	0.026	0.673	−0.063	0.310	−0.118	0.058	−0.055	0.378
carbonated soft drinks	0.063	0.316	−0.009	0.888	0.009	0.886	−0.025	0.691	−0.096	0.124	−0.057	0.364
light drinks (with reduced energy value)	0.074	0.232	0.041	0.514	0.018	0.768	0.067	0.280	−0.019	0.762	0.116	0.063
carbonated mineral water	0.028	0.655	0.182	**0.003**	0.087	0.162	−0.068	0.274	−0.026	0.673	0.048	0.441

WC, waist circumference; SBP, systolic blood pressure; DBP, diastolic blood pressure; TG, triglycerides; HDL-C, high-density lipoprotein cholesterol; GLC, glucose; r_s_, Spearman’s rank correlation coefficient. Statistically significant: *p* < 0.05 (bold font represents statistically significant *p*-values).

## Data Availability

The data presented in the study are available on request from the corresponding author. The data are not publicly available due to privacy restrictions.

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
