# Peer review of "Metabolic Syndrome and Dietary Habits in Hospitalized Patients with Schizophrenia: A Cross-Sectional Study"

_medicina, 2021, doi:10.3390/medicina57030255_

Round 1

Reviewer 1 Report

This cross-sectional observational study aims to determine the prevalence of metabolic syndrome in hospitalized patients with schizophrenia and to investigate the differences in socio-demographic, clinical, and lifestyle distinguish patients with and without metabolic syndrome.

The paper is adherent to the aims of the journal and his results are useful for clinicians (to improve the nutritional support for their patients) and researchers (to further investigate the relationship between schizophrenia and metabolic syndrome). However, the paper needs major revisions.

Abstract

Authors need to report the limitations of the study in abstract

Materials and Methods

This section is high problematic.

The sentences “In total, 259… 25% female patients).” (lines 111-114) should be shift in Results section.

The sentences “The needed sample … poorer dietary habits.” (lines 115-127) should be shift in Statistical analysis section.

The sentence “Out of 259 … (table 2)” (lines 136-139) should be delete (the information is in table 2)

Most of the information in section 2.3 Outcome measures correspond to the assessment.

Authors could cut some non-essential technical information included in lines 152-178 (for example “using digital medical stadiometric…”, or use the aneroid…”).

Results

To facilitate the manuscript reading the Authors should report in the text only the results of total sample and the significant differences between groups (see lines 237-242 and 267-275). The complete analysis of the differences between the groups could stay at the tables.

Authors could cut Fig.1 because the same information are in the text.

Discussion

The findings of the study show the absence of a relationship between metabolic syndrome and dietary habits. Authors should discuss alternative hypothesis (genetic, ...) explain the high rate of metabolic syndrome in their patients.

Authors should report your opinion on the (unusual) absence of a relationship between metabolic syndrome and physical activity or typical/atypical antipsychotic use.

Reviewer 2 Report

Review medicina-1124152

In a descriptive study with a cross-sectional study design, the authors qualitatively mapped the eating habits of 259 people with schizophrenia admitted to a psychiatric hospital and compared them in people with (n = 124) and without (n = 135) metabolic syndrome according to the criteria of Joint Interim Statement (JIS). This is an interesting study, but this referee also has some important reservations.

The authors use a less common definition for the metabolic syndrome. It is useful to compare these criteria in a table with other criteria, especially those of the International Diabetes Federation IDF. It should not be too difficult to indicate on the basis of which criteria patients of the current study are more likely to have metabolic syndrome than if the IDF criteria had been applied. Figure 1b is not very informative (the most important data is already in the text). It is better replaced by a bar graph showing the number of patients meeting the individual criteria of the JIS. This ties in with the description of the last part of the results section. According to the authors' statement, a significant proportion of patients (n = 80) are not legally competent. This may be something special for this country, but usually people with schizophrenia are considered essentially legally competent (even if they stay in hospital involuntarily) and seeking proxy consent is estimated undesirable (being patronizing). The reasons for deviating from this (for example because it was demanded by the ethics committee or the presence of other problems such as dementia) must be explicitly described. Did the patients' legal status affect their dietary freedom?

The type of ward where patients stayed and the length of their hospital stay (≤ 1yr vs. longer) can influence their diet. Shortly after admission, patients will not have adjusted their diet yet. Would you like to classify the type of ward according to the extent to which patients provide their own food? Are there differences in diet as patients have more freedom to take care of their own food and how does this relate to the development of metabolic syndrome?

Some minor points are:

Line 162: An accuracy of 0.1 cm is unrealistic.

Line 240: The reviewer is surprised that more than half of the participants are retired. Please discuss this further and put it into context in Section 4.

Line 247: Classifying the participants according to their original socio-economic backgrounds can be a useful addition.

Line 260: This table is currently not very informative and can perhaps go to the supplementary section and be replaced by a graph/table with correlation data.

Line 338: It is not possible to indicate that eating habits are inadequate without discussion on which this statement is based. What diet is customary and adequate for your country?

Line 377: The degree of deviation from the meals provided by the hospital is essential for this study. Please substantiate this in the results section and discuss this in more detail here.

Round 2

Reviewer 1 Report

Authors significantly improved the manuscript after revision. 

Reviewer 2 Report

No comments anymore. Very nice paper, congratulations